# Comprehensive Gene Expression Analyses of Immunohistochemically Defined Subgroups of Muscle-Invasive Urinary Bladder Urothelial Carcinoma

**DOI:** 10.3390/ijms22020628

**Published:** 2021-01-10

**Authors:** Bohyun Kim, Insoon Jang, Kwangsoo Kim, Minsun Jung, Cheol Lee, Jeong Hwan Park, Young A. Kim, Kyung Chul Moon

**Affiliations:** 1Department of Pathology, Seoul National University College of Medicine, Seoul 03080, Korea; cherish1107@naver.com (B.K.); jjunglammy@gmail.com (M.J.); fejhh@hanmail.net (C.L.); 2Biomedical Research Institute, Seoul National University Hospital, Seoul 03080, Korea; isjang1324@gmail.com; 3Transdisciplinary Department of Medicine & Advanced Technology, Seoul National University Hospital, Seoul 03080, Korea; kksoo716@gmail.com; 4Department of Pathology, Seoul Metropolitan Government-Seoul National University Boramae Medical Center, Seoul 03080, Korea; hopemd@hanmail.net (J.H.P.); pathgirl@daum.net (Y.A.K.); 5Kidney Research Institute, Medical Research Center, Seoul National University College of Medicine, Seoul 03080, Korea

**Keywords:** muscle invasive bladder cancer, molecular classification, cytokeratin 5/6, cytokeratin 20

## Abstract

A number of urinary bladder urothelial carcinoma (UB UC) mRNA-based classification systems have been reported. It also has been observed that treatment response and prognosis are different for each molecular subtype. In this study, cytokeratin (CK)5/6 and CK20 immunohistochemistry (IHC) were performed, and IHC-based subgroup classification was applied. UB UC was classified into CK5/6 single-positive (SP), CK20 SP, double-positive (DP) and double-negative (DN) subgroups, and transcriptional analysis was performed. The results of gene ontology (GO) terms and functional analysis using differentially expressed genes indicate that, CK5/6 SP and DP subgroups were enriched in cell migration, immune activation, interleukin 6-Janus kinase-signal transducer and activator of transcription 3 (IL6-JAK-STAT3) signaling pathway and tumor necrosis factor-α signaling via the nuclear factor-κB (NF-κB) signaling pathway signature gene. In addition, compared with the other subgroups, the DN subgroup showed inhibited cell movement, cell migration, and cell activation. Furthermore, in survival analysis, the CK5/6 SP subgroup was significantly associated with poor progression-free survival (*p* = 0.008). The results of our study indicate that the CK5/6 positive subgroup exhibited high gene expression signature related to aggressive behavior and exhibited worse clinical outcome.

## 1. Introduction

Bladder cancer worldwide is the 10th most common form of malignant tumor and has the 14th highest cancer-associated mortality [1]. Urinary bladder urothelial carcinoma (UB UC) is the most common tumor found in the urinary bladder. UB UC can be divided into papillary urothelial carcinoma, which shows papillary growth, and invasive urothelial carcinoma, which initially shows invasive growth. The two types of tumor are known to have different pathogenesis. Recent studies reported that papillary urothelial carcinoma originates from the intermediate cells of the urothelium and that carcinoma in situ and invasive urothelial carcinoma originate from the basal cells of the urothelium [2].

Recently, several studies have been published that utilized next-generation sequencing (NGS) analysis to analyze and classify UB UC according to gene expression [3,4,5]. While several different mRNA-based subtype classifications exist, some of them overlap with one another. Universally, UB UC is classified into luminal and basal subtypes, and according to each classification system, it is further classified into Tp53-like type, urothelial-like A type, infiltrated type, genomically unstable type, mesenchymal-like type, and small-cell/neuroendocrine-like type [3]. Each subtype shows high expression of specific genes. While the basal type shows high expression of stem cell or basal urothelial cell markers such as KRT14, KRT5, KRT6, and CD44, the luminal type shows high expression of urothelial differentiation markers such as KRT20, GATA3, and FOXA1 [3,5]. Furthermore, consistent with gene expression in each subtype, on immunohistochemistry (IHC) staining, the basal type shows high expression of cytokeratin (CK)5/6, and the luminal type shows high expression of CK20 [3,5]. Through the consensus meeting, bladder cancers showing positive KRT5/6, KRT14 expression and negative GATA3, FOXA1 expression were classified into a specific subtype, known as basal-squamous-like (BASQ) [6]. In the previous study, we classified muscle-invasive urinary bladder cancer (MIBC) into BASQ or non-BASQ subtypes based on the IHC results [7].

Molecular subtypes for UB UC have been established recently, and studies attempting to demonstrate their clinical significance are actively in progress [8]. Molecular subtype classification in UB UC also has the potential to play a major role in treatment decisions and prognosis predictions [9,10]. In fact, various studies have reported that prognosis, neoadjuvant chemotherapy response, and driver gene mutation vary among molecular subtypes in UB UC. Neoadjuvant cisplatin-based chemotherapy is the standard of care for high-risk MIBC. Neoadjuvant chemotherapy response varies among with molecular subtype, especially the p53-like subtype, which is known to be chemo-resistant, and basal-type MIBC, which is reported to benefit from neoadjuvant chemotherapy because of its chemo-sensitive nature [3,11]. In a phase II clinical trial of IMvigor210, different treatment responses to the immune check point inhibitor atezolizumab and prognosis were reported among molecular subtypes in patients with locally advanced urothelial carcinoma. In phase II clinical trial of CheckMate 275 using nivolumab, another immune check point inhibitor, different treatment responses were reported among molecular subtypes in patients with advanced stage urothelial carcinoma [12,13,14,15]. Thus, molecular subtype classification of UB UC is crucial for effective prognosis estimation and treatment planning.

In the mRNA-based subtype classification, high expression of KRT5 is classified as the basal type and high expression of KRT20 is classified as the luminal type [4,5,16,17]. Accordingly, with IHC, the basal type shows high expression of CK5/6 and the luminal type shows high expression of CK20 [3,5]. However, little has been published regarding cases in which both CK5/6 and CK20 showed high expression or in which neither protein showed high expression. In this study, two additional groups besides the basal and luminal types have been defined. These two groups are the double-positive type and double-negative type, for which we will investigate the molecular genetic characteristics. We could expect great clinical utility if two IHC assays could predict the molecular genetic characteristics of a tumor.

## 2. Results

### 2.1. Subgroup Classification of RNA Sequencing Group

Thirty cases of MIBC in patients who underwent a radical cystectomy between 2016 and 2018 and for whom we were able to acquire fresh frozen tissues were included. With these samples, we subdivided the cases into four subgroups based on CK5/6 and CK20 IHC expression. The CK5/6 single-positive (CK5/6 SP) subgroup was $CK5/6^{high}$/$CK20^{low}$, the CK20 single-positive (CK20 SP) subgroup was $CK5/6^{low}$/$CK20^{high}$, the double-positive (DP) subgroup was $CK5/6^{high}$/$CK20^{high}$ and the double-negative (DN) was $CK5/6^{low}$/$CK20^{low}$. Six cases of the CK5/6 SP subgroup, 10 cases of the CK20 SP subgroup, nine cases of the DP subgroup and five cases of the DN subgroup were identified (Figure 1). 

Three cases were selected from each subgroup, and mRNA sequencing was performed on fresh frozen tissue samples of 12 patients. The mean age of patients was 68.7 years (range, 55–83) at diagnosis, and the male-to-female sex ratio was 10:2. Seven patients were in stage IIIA and five patients were in stage IIIB according to the 8th edition of the TNM staging system of the American Joint Committee on Cancer (AJCC) [18]. Lymph node metastasis was found in six cases. All 12 cases were classified as high grade according to the World Health Organization/ International Society of Urologic Pathologists (WHO/ISUP) grading system [19].

We additionally performed CD44, CK14, GATA3, FOXA1 and P53 IHC in 12 cases. In CD44, a basal-type marker, IHC expression was positive in all CK5/6 SP and DP subgroup cases and mostly negative in CK20 SP and DN subgroups. In CK14, another basal-type marker, IHC expression was positive in one case in the DP subgroup and two cases in the CK5/6 SP subgroup and was negative in all remaining cases. In GATA3, a luminal-type marker, IHC expression was positive in all three CK20 SP subgroup cases and negative in all three CK5/6 SP subgroup cases. FOXA1 IHC expression was positive in two cases in the CK20 SP subgroup and one case in the DN subgroup and was negative in all remaining cases, including CK5/6 SP and DP subgroup cases. P53 IHC expression showed a wide range from negative to greater than 95% positive expression.

### 2.2. Differential Expression Genes (DEGs) between Each Subgroups

RNA sequencing data were analyzed from 12 MIBC tissues. Using adjusted *p*-value < 0.05 and |fold change| ≥2 as the cut-offs, we identified 38 differential expression genes (DEGs) between the DP and CK20 SP subgroups, 98 DEGs between the DP and CK5/6 SP subgroups, 433 DEGs between the DP and DN subgroups, 183 DEGs between the CK20 SP and CK5/6 SP subgroups, 256 DEGs between the CK20 SP and DN subgroups, and 614 DEGs between the CK5/6 SP and DN subgroups. In total, 1062 DEGs were identified. Compared with the DP, CK5/6, and DN subgroups, the CK20 SP subgroup had 10, 77, and 174 upregulated genes, respectively, and 28, 106 and 82 downregulated genes, respectively. Compared with the DP and CK 20 SP and the DN subgroups, the CK5/6 SP subgroup had 43, 106, and 429 upregulated genes, respectively, and 55, 77, and 185, downregulated genes, respectively. Compared with the DP, CK20 SP and CK5/6 SP subgroups, the DN subgroup had 79, 82 and 185 upregulated genes, respectively, and 354, 174 and 429 downregulated genes, respectively. A Venn diagram of DEGs in the four major comparison conditions is shown in Figure 2.

### 2.3. Gene Ontology (GO) Analysis

The results of Gene Ontology (GO) analysis between each subgroup are summarized in Table 1. Between the CK5/6 SP and CK20 SP subgroups, the leukocyte aggregation GO term was identified. Some related DEGs (S100A9, CD44 and S100A8) were upregulated in the CK5/6 SP subgroup. Between the CK5/6 SP and DN subgroups, various GO terms related to immune response and the tumor necrosis factor (TNF) signaling pathway were identified, and various related DEGs (CD86, HLA-DMB, CD209, CCL19, TLR1, BTK, IRAK3 and TLR 6) were upregulated in the CK5/6 SP subgroup. Between the DP and CK20 SP subgroups, GO terms related to immune response were identified, and various related DEGs (CCL4L1, S100A9, IL1B and LILRB1) were upregulated in the DP subgroup. Between the DP and DN subgroups, GO terms related to cell proliferation, immune response, mitogen-activated protein kinase (MAPK) signaling pathway and TNF signaling pathway were identified, and various related DEGs (CCL14, CD74, CD4, CD86, FLT3 and LRRK2) were upregulated in the DP subgroup.

When comparing the CK5/6-positive and CK5/6-negative groups, we identified 226 DEGs. Compared to the CK5/6-negative group, the CK5/6-positive group had 169 upregulated genes and 57 downregulated genes. The CK5/6-positive group was enriched with cell migration, immune response, MAPK signaling pathway and TNF signaling pathway associated GO terms. Compared to CK5/6- negative tumors, many DEGs (ACVR1, CSF1R, SEMA4A, SASH3, CD74, HLA-DMB, TLR1, BTK, IRAK3, MAP4K1 and CD40) associated with previously mentioned GO terms were upregulated in CK5/6-positive group tumors (Table 2).

### 2.4. Ingenuity Pathway Analysis (IPA) and Gene Set Enrichment Analysis (GSEA)

The ingenuity pathway analysis (IPA) results showed that some functions related to immune response and cell migration were activated in the CK5/6 SP and DP subgroups (Table 3). Regulator effect analysis also indicated analogous results (Appendix A). The significant ingenuity canonical pathways of DEGs between four subgroups are listed in Figure 3.

Gene set enrichment analysis (GSEA) confirmed that the interleukin 6-Janus kinase-signal transducer and activator of transcription 3 (IL6-JAK-STAT3) signaling pathway, inflammatory response and TNF-α signaling via nuclear factor-κB (NF-κB) signaling pathway were significantly enriched in the DP and CK5/6 SP subgroups. (Figure 4) However, in the comparison conditions of the CK5/6 SP and CK20 SP subgroups, related functions did not exhibit a significant difference.

### 2.5. Expression of Gene Signature Markers

We compared the expression patterns of gene signature markers between four subgroups (Appendix A). Expression of basal-type markers was most enriched in the CK5/6 SP subgroup, and the DP, CK20 SP, and DN subgroups followed in respective order. In contrast, expression of luminal-type markers was most enriched in the CK20 SP subgroup, and the DP, DN, and CK5/6 SP subgroups followed in respective order. Expression of p63-associated genes was the highest in the CK5/6 SP subgroup followed by the DP subgroup, and the CK20 SP and DN subgroups showed low expression. Expression of TP53-like signature genes was lowest in the DN subgroup. Expression of immune cell-associated genes was highest in the DP subgroup. Compared with the CK5/6-negative group, expression of epithelial-mesenchymal transition (EMT) markers was enriched in the CK5/6-positive group. Expression of cell adhesion markers was enriched to the highest degree mostly in the CK5/6 SP subgroup, which was followed by the DP subgroup. Expression was relatively low in the remaining CK5/6-negative group. Expression of TNF and MAPK signaling pathway was enriched in the DP and CK5/6 SP subgroups. The DN subgroup showed the lowest expression.

### 2.6. Clinicopathological Analysis of Immunohistochemistry (IHC)-Based Subgroups

The clinicopathological and demographic characteristics of the 189 patients for whom we performed IHC analysis are summarized in Table 4. Overall, 189 patients were included in this study, including 158 men and 31 women. The age of the patients ranged from 37 to 87 years with a mean age of 68 years. According to the 8th edition of the TNM staging system of the AJCC, 172 patients were in pT2, 10 patients were in pT3, and 7 patients were in pT4. According to the WHO/ISUP grading system, seven cases were classified as low grade, and 182 cases were classified as high grade. Only conventional UB UC cases were selected excluding specific variant cases. IHC was performed to subdivide subgroups, and the results were as follows. There were 61 cases in the CK5/6 SP subgroup, 13 cases in the DP subgroup, 70 cases in the CK20 SP subgroup and 45 cases in the DN subgroup that were confirmed. (Table 5).

The follow-up period ranged from 1 to 277 months, the median follow-up period was 16 months, and the median survival period was 92 months. Of a total of 189 patients, clinical follow-up data for 185 patients were available. During the follow-up period, disease progression was found in 95 cases, and death occurred in 128 cases.

We performed the Kaplan–Meier survival analysis according to IHC-based classification. Of the four IHC based subgroups, the CK5/6 SP subgroup had the worst progression-free survival (PFS) (*p* = 0.008). The Kaplan–Meier analysis also showed that the high expression of CK5/6 was associated with unfavorable PFS (*p* = 0.005) (Figure 5). In contrast, low expression of CK20 was associated with unfavorable PFS (*p* = 0.028). However, there was no significant difference in overall survival (OS) according to IHC-based classification (*p* = 0.709) or IHC expression (*p* = 0.840, CK5/6; *p* = 0.286, CK20).

## 3. Discussion

In recent years, research on molecular-based UB UC subtype classification has been actively underway. As gene expression profiles were reported to show different disease progressions, responses to chemotherapy and survival rates, the UB UC molecular subtype classification became a necessary step in UB UC diagnosis and treatment planning. In MIBC, the basal type is known to arise from basal and stem cells of normal urothelium, and the luminal type is known to arise from terminally differentiated superficial umbrella cells [20]. During urothelium differentiation, basal and intermediate cells express KRT5 but do not express KRT20. Furthermore, terminal differentiation process is associated with suspension of KRT5 expression and start of KRT20 expression [21]. Additionally, as KRT14 is also involved in this process, basal cells are known to show KRT14+KRT5+KRT20−, intermediate cells show KRT14−KRT5+KRT20−, and differentiated cells show KRT14−KRT5−KRT20+ [22]. In MIBC, the basal subtype shows high expression of KRT5, KRT6, KRT14, CD44 and CDH3, shows chemo-sensitive properties, is intrinsically aggressive, and is associated with poor prognosis. The luminal subtype shows high expression of UPK, KRT20, FOXA1, and GATA3, shows chemo-resistant properties, and seems to be less aggressive [20]. In addition to mRNA levels, IHC protein level markers include CK5/6 and CK14 for the basal subtype and CK20, GATA3, and Uroplakin2 for the luminal subtype. Furthermore, one study indicated that expression of CK5/6 and CK20 was inversely related in MIBC [3]. Based on these results, in a practical context, affordable IHC antibodies, CK5/6 and CK20, were selected as surrogate markers for IHC-based subgroup classification.

Previous studies utilized NGS analysis to assess gene expression patterns and reported differences in CK expression patterns between molecular subtypes. In this study, we took a different approach in case selection. Cases of the CK5/6 SP, CK20 SP, DP and DN subgroups were selected based on CK IHC results, and gene expression profiles were evaluated with these cases. Furthermore, based on IHC, CK5/6 and CK20 protein expression areas were accurately evaluated before tissue collection, which led to more precise tumor tissue collection.

MIBC cases with sufficient fresh-frozen tissues were included, which resulted in 30 samples classified to four subgroups based on CK5/6 and CK20 IHC results. Nine cases in the DP subgroup and five cases in the DN subgroup among 30 cases may indicate a relatively high proportion of DP and DN subgroups in UB UC molecular subtype classification. The DP and DN subgroup cases may be in a stage of transition between molecular subtypes or they may be subtypes completely independent from CK5/6 SP and CK20 SP subgroups. However, research regarding this aspect of subtype classification has not been presented to date. Compared to the CK5/6 SP and CK20 SP subgroups, the DP subgroup showed a relatively high expression of basal- and luminal-type markers, which gave us the impression of a mixed phenotype. This subgroup showed the strongest immune signature genes expression. However, in GO, IPA, and GSEA functional analysis, the DP subgroup did not show a significant difference from the CK5/6 SP subgroup. Based on these results, the DP subgroup was expected to be close to the CK5/6 SP subgroup in tumor characteristic aspects. The DN subgroup, in every comparison with the three other subgroups, showed the highest number on GO and Kyoto Encyclopedia of Genes and Genomes (KEGG) pathways with significant differences. The DN subgroup also showed mostly low gene expression on biological signature gene cluster expression analysis. In GO, IPA, and GSEA functional analysis, the DN subgroup showed no significant difference from the CK20 SP subgroup, except for “activation of cells’ associated molecules”, in the IPA result. The DN subgroup is the most similar to the CK20 SP subgroup among the three other subgroups, but we think it is a unique subtype that shows differences in gene expression from the other subgroups.

Expression of p63-associated genes was high in the CK5/6 SP and DP subgroups. TP63, which is a transcription factor associated with basal/stem cells in the urothelium, is known to be activated in the basal-type MIBC and to regulate basal gene expression signature [3]. In line with previous studies, which reported that p63-associated genes were enriched in the basal-type MIBC, CK5/6 SP and DP subgroups showed high expression of p63-associated genes.

In comparison between subgroups, expression of TP53-like signature genes was the lowest in the DN subgroup and varied among cases. Six cases, which showed diffusely strong positive results on IHC, showed high expression of TP53-associated gene mRNA. Association between TP53 mRNA expression and TP53 IHC protein expression showed a positive correlation.

Expression of immune response-associated genes was high in the DP subgroup, and the CK5/6 SP, DN, and CK20 SP subgroups followed in respective order. Dividing the genes into two groups based on CK5/6 IHC results, immune response GO term associated DEGs were more enriched and associated genes were more upregulated in DP and CK5/6 SP subgroups than in DN and CK20 SP subgroups. Additionally, on IPA analysis, immune activation was more upregulated in DP and CK5/6 SP subgroups than in the other subgroups. In a previous study in our group, it was found that programmed cell death-ligand 1 (PD-L1) expression was enriched in the BASQ subtype positive for CK5/6 and CK14 on IHC [7]. These results correlate with previous studies, which reported high immune gene signature expression in the basal subtype [8,23].

Expression of EMT marker was enriched in the DP and CK5/6 SP subgroups. The EMT is the process of epithelial or endothelial cells acquiring mesenchymal phenotypes, and is associated with tumor progression and metastasis process [24,25,26,27]. Furthermore, some studies reported that, in various cancers, EMT showed a strong correlation with immune activation, and had high expression of programmed cell death-1 (PD-1), PD-L1, CTLA4, OX40L and PD-L2. In these studies, activation of the immune cell signaling pathway was observed in the EMT setting, which was interpreted as an indication that the EMT may facilitate a change of the tumor microenvironment [28,29]. Additionally, Chen et al. suggested that microRNA-200 (miR-200) formed a negative feedback loop with ZEB1, an EMT activator, suppressed the EMT, and suppressed PD-L1 expression. These authors demonstrated that the EMT is linked to the immunosuppression through the miR-200/ZEB1 axis [30,31]. The miR-200 family is known to be enriched in the luminal type, which also correlates well with the results of this study, as CK20 SP and DN subgroups showed low EMT and immune response-associated gene expression [3].

Expression of cell adhesion markers showed low expression of tight junction-related genes TPJ2 and CLDN4 and high expression of desmosome-related genes DSC3 and PKP1, gap junction-related genes GJB3 and GLB4, and epithelial integrin genes ITGA6 and ITGB4 in the CK5/6 SP subgroup. These results imply that the CK5/6 SP subgroup shows increased expression of cell adhesion, especially basolateral cell adhesion-related genes. This result is in keeping with previous studies, which reported high cell adhesion gene expression signature in the mRNA-based basal subtype, including urobasal B and SCC-like subtype [16].

Expression of the MAPK and TNF signaling pathways was enriched in the DP and CK5/6 SP subgroups. Most of the DEGs associated with these signaling pathway were upregulated in the DP and CK5/6 SP subgroups.

On GSEA analysis, the IL6-JAK-STAT3 signaling pathway and TNF-α signaling via NF-κB signaling pathway were significantly enriched in DP and CK5/6 SP subgroups. In contrast, the DN subgroup was found to be less sensitive to this pathway. The IL6-JAK-STAT3 signaling pathway is upregulated in various types of cancer and its hyperactivation is associated with adverse clinical outcome [32,33,34]. Furthermore, the TNF-α/NF-κB signaling pathway is known to be involved in the tumor invasion and metastasis [35,36]. Considering the GSEA results, the CK5/6-positive subgroup (CK5/6 SP and DP subgroups) is a more aggressive subtype than the CK5/6-negative or at least the DN subgroup.

On IPA analysis, compared to the other subgroups, the DN subgroup showed downregulated cell movement, cell migration, and cell activation. Such cellular functions, which are associated with cell motility, are closely related to cancer invasion and cancer metastasis [37,38,39,40]. Based on IPA analysis we can expect the DN subgroup to have a less aggressive behavior than the three other subgroups.

Different molecular subtypes were reported to have different prognoses and chemotherapy sensitivity in MIBC. In addition, we also found that the IHC-based classification has prognostic value. Compared with the CK5/6-negative subgroup (CK20 SP and DN subgroups), the CK5/6-positive subgroup (CK5/6 SP and DP subgroups), which implies the high EMT, IL6-JAK-STAT3 signaling pathway, TNF-α/NF-κB signaling pathway and immune gene expression signature, show adverse disease progression outcome, and this result is in keeping with the findings of previous studies [3,5]. It is assumed that the effective immune escape process that arises by expression of PD-L1 may be related to disease progression. Although the DP subgroup showed no difference in disease progression rate from other CK20 SP and DN subgroups, this consequence is considered because there were very few cases of the DP subgroup in this study population. In contrast to the PFS, there was no significant difference in OS rate between IHC-defined subgroups. The difference in PFS for each subgroup but no difference in OS can be interpreted as meaning that the post-progression period is considerably long. We think this is related to the fact that most of the subjects of this study consist of pathologic stage T2 cases. Since it was composed of cases with a good overall prognosis, there was no significant difference in the OS analysis for each subgroup.

In the personalized precision medicine era, subgrouping of patients based on molecular characteristics has considerable influence on selecting therapeutic regimens and forecasting therapeutic responses and prognoses. In addition, mRNA sequencing analysis is a good method to examine phenotype characteristics of a tumor, however, in this study we utilized an IHC-based classification that is simpler and more affordable. We examined the molecular characteristics of subgroups based on IHC classification. In summary, we found that the CK5/6-positive subgroup showed a high gene expression signature related to aggressive behavior and showed worse clinical outcomes.

## 4. Materials and Methods

### 4.1. Tissue Samples and Case Selection

In total, MIBC tissues from 30 patients who underwent a radical cystectomy at Seoul National University Hospital (SNUH) from 2016 to 2018 were included in this study. First, 30 fresh frozen tissue samples from 2016 to 2018 were investigated. The section containing prominent tumor tissue was selected, of which one section was prepared as a formalin-fixed paraffin-embedded (FFPE) tissue block, and the corresponding symmetric section was fresh frozen with liquid nitrogen and preserved in −70 °C until tumor excision. A total of 30 samples were classified to four subgroups based on IHC results for CK5/6 and CK20. In each subgroup three representative cases were selected, and RNA sequencing was carried out with a total of 12 fresh frozen tissue samples. All 12 patients were included in the prospective studies. Written informed consent was obtained from all patients enrolled in this mRNA analysis group.

Additionally, 189 MIBC patients who underwent transurethral resection of the bladder or radical cystectomy at SNUH or Seoul Metropolitan government-Seoul National University Boramae Medical Center were included in the survival analysis group. A total of 189 FFPE block tissue samples from 2004 to 2010 were investigated. A tissue microarray (TMA) block was prepared from FFPE tissue blocks (SuperBioChips Laboratories, Seoul, Republic of Korea). Two cores (2 mm in diameter) containing invasive tumor areas were obtained from each case. This study was approved by the Institutional Review Board (IRB) of SNUH (IRB No C-1701-083-823, 24 January 2017).

### 4.2. Immunohistochemistry

IHC for CK5/6, CK20, CK14, GATA3, FOXA1, CD44 and TP53 was performed with an automatic immunostainer (BenchMark XT; Ventana Medical Systems, Tucson, AZ, USA), following the manufacturer’s instructions. The primary antibodies against CK5/6 (1:100; D5/16 B4; Dako, Glostrup, Denmark), CK20 (1:50; Ks 20.8; Dako), CK14 (1:300; LL002; Cell Marque, Rocklin, CA), CD44 (1:100; 156-3C11; Thermo Fisher), GATA3 (L50-823; 1:500; Cell Marque), FOXA1 (1:500; PA5-27157; Thermo Fisher, Waltham, MA, USA) and p53 (1:1000; DO7; Dako) were used. Immunohistochemical positive control tissues were tonsil tissue for CK5/6 and CD44, duodenal mucosa epithelium for CK20, squamous cell carcinoma tissue for CK14, urothelial carcinoma tissue for GATA3, gastric mucosa epithelium for TP53, and breast cancer tissue, previously confirmed with positive expression for FOXA1.

The full-section IHC staining was performed on all 12 cases, and in each case, two different invasive tumor areas were investigated. Additionally, the same IHC was carried out for 189 MIBC tissue TMA blocks.

The evaluation results of CK5/6, CK20, CK14, GATA3 and FOXA1 IHC were derived from a previously published study [7]. Briefly explaining the evaluation criteria again, >20% expression was defined as high expression status. In previous studies, 20% cut-off value was reported to be ideal for classification of molecular subtypes in bladder cancer [41,42]. The same cut-off was applied for CK5/6, CK20, CK14 and CD44. For GATA3 and FOXA1, the percentage of nuclear stained cells and staining intensity were considered. Staining intensity was scored from 0 to 3+ (0: no staining, 1+: weak staining, 2+: moderate staining, and 3+: strong staining). IHC staining with both 3+ intensity and more than 20% of tumor cells was defined as high expression. Two pathologists (B.K. and C.L.) evaluated IHC staining at two different time points, without awareness of the previous results at the second evaluations. In the case of discrepant results between evaluations another pathologist (K.C.M) was consulted before making the final decision.

### 4.3. RNA Sequencing

After IHC results were identified on the FFPE tissue block, mRNA was extracted from the corresponding location of the symmetric fresh-frozen MIBC tissues. A 3-mm-sized plunger was used for punching out the fresh-frozen tissue. RNA library was assembled with TruSeq RNA Access Library Prep Kit, and base sequence analysis was carried out with Illumina HiSeq 2500 platform (Illumina, San Diego, CA, USA). (Macrogen, Inc., Seoul, Korea) Paired demultiplexed fastq files were generated, and initial quality control was performed using FastQC (Phred quality score >30). The adaptor sequences were removed by using Trimmomatic program and trimmed data were mapped to the reference genome (UCSC hg19) using the HISAT2 and Bowtie2. Previously known gene/transcripts were assembled with the StringTie program. Raw data were normalized and, mRNA expression data were presented as reads transcript per million and were transformed into log 2 volume values for the analysis.

### 4.4. Functional Analysis

In this study, DEGs were identified with DESeq2. Statistical analysis was performed on selected genes whose median count for each gene was greater than 5 in at least one comparison combination. Functional analysis of DEGs was performed using GO and the KEGG pathway (adjusted *p*-value < 0.05 and |fold change| ≥2). The analyses were performed by biological process, molecular function, and cellular component, which are the GO subcategories. Functional annotation based on the KEGG database was implemented. Additionally, the functional analyses were performed with the use of IPA.

### 4.5. Statistical Analysis

The association between IHC defined subgroups with immunohistochemistry expression was evaluated by the chi-squared test. The associations between IHC based subgroups and PFS, and OS were evaluated by the Kaplan–Meier method with the log-rank test. Statistical analyses were performed using SPSS software (version 23; IBM, Armonk, NY, USA). Two-sided *p*-values < 0.05 were considered to be statistically significant.

## Figures and Tables

**Figure 1 ijms-22-00628-f001:**
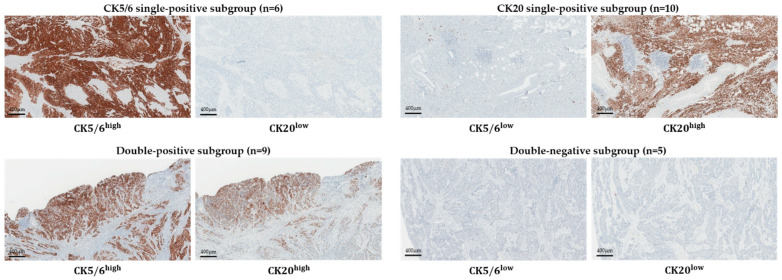
Subgrouping of muscle-invasive urinary bladder cancer (MIBC) by CK5/6 and CK20 immunohistochemistry (IHC).

**Figure 2 ijms-22-00628-f002:**
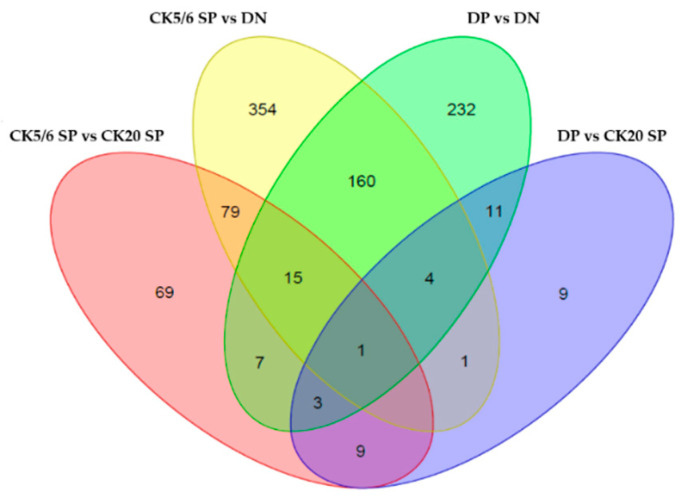
Venn diagram of differential expression genes (DEGs) in four major comparison conditions.

**Figure 3 ijms-22-00628-f003:**
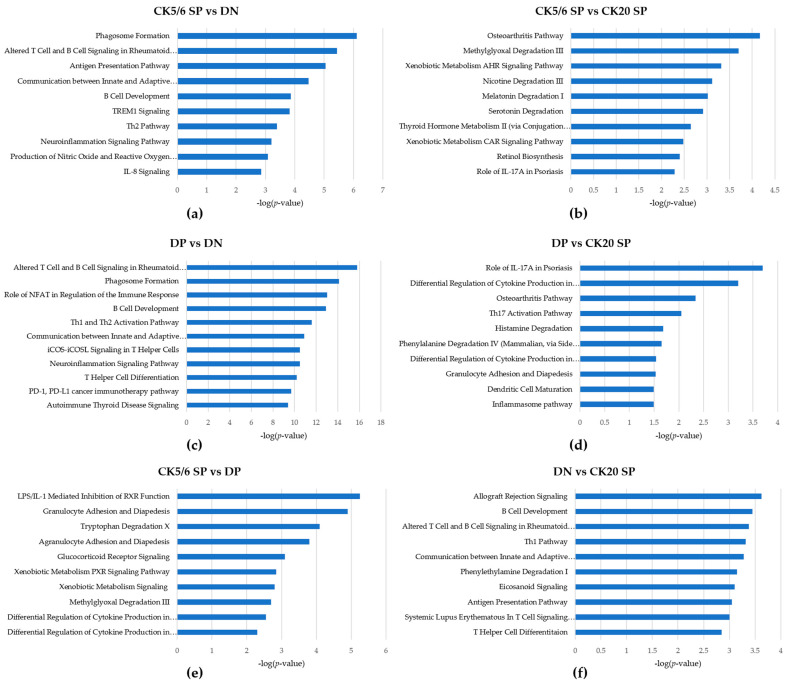
Ingenuity canonical pathways of DEGs between four subgroups: (**a**) CK5/6 SP vs. DN; (**b**) CK5/6 SP vs. CK20 SP; (**c**) DP vs. DN; (**d**) DP vs. CK20 SP; (**e**) CK5/6 SP vs. DP; (**f**) DN vs. CK20 SP.

**Figure 4 ijms-22-00628-f004:**
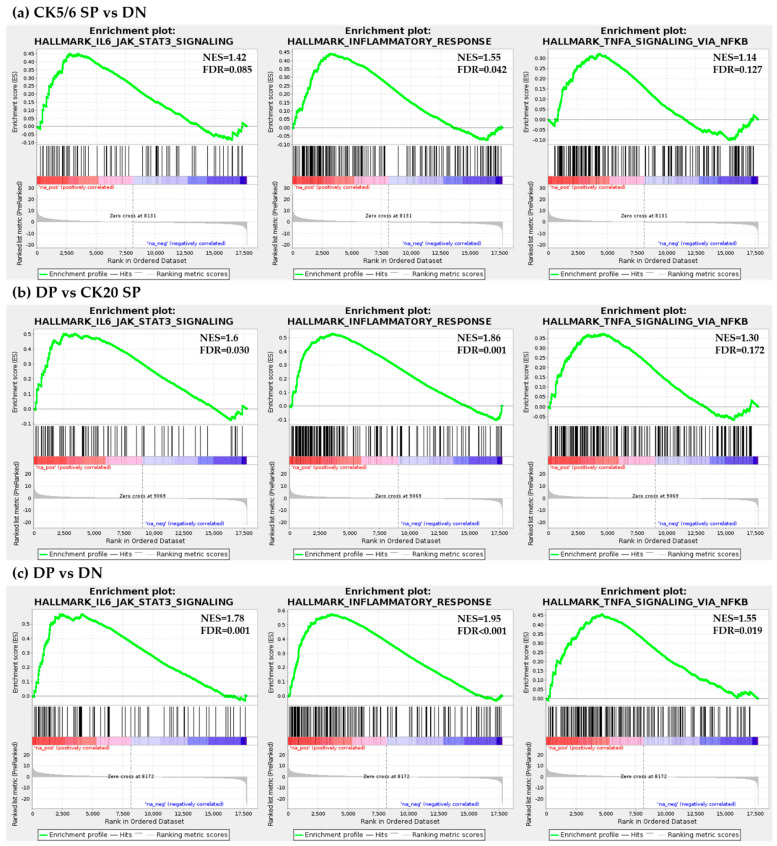
Gene set enrichment analysis results: (**a**) CK5/6 SP vs. DN; (**b**) DP vs. CK20 SP; (**c**) DP vs. DN.

**Figure 5 ijms-22-00628-f005:**
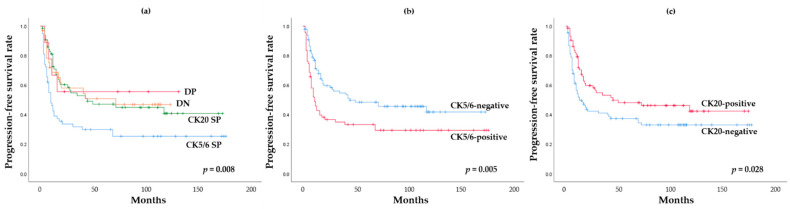
Progression free survival analysis: (**a**) impact of IHC-based classification; (**b**) impact of CK5/6 expression; (**c**) impact of CK20 expression.

**Table 1 ijms-22-00628-t001:** The Gene Ontology (GO) analysis results between each subgroup.

CK5/6 single-positive (CK5/6 SP) vs. double-positive (DP)
None related to major cellular function
CK5/6 SP vs. CK20 single-positive (CK20 SP)
Immune response (leukocyte aggregation)
CK5/6 SP vs. double-negative (DN)
Immune response (positive regulation of lymphocyte proliferation, positive regulation of T cell activation, positive regulation of T cell proliferation, regulation of B cell receptor signaling pathway, neutrophil activation involved in immune response, and positive regulation of leukocyte cell-cell adhesion)
Tumor necrosis factor (TNF) signaling pathway (MyD88-dependent toll-like receptor signaling pathway and positive regulation of NF-κB transcription factor activity)
DP vs. CK20 SP
Immune response (regulation of T cell proliferation and positive regulation of inflammatory response)
DP vs. DN
Cell proliferation (positive regulation of cell proliferation)
Immune response (T cell proliferation, regulation of T cell activation, regulation of immune response, regulation of inflammatory response, regulation of B cell proliferation, positive regulation of lymphocyte proliferation, positive regulation of T cell activation, and positive regulation of T cell proliferation)
TNF signaling pathway (positive regulation of tumor necrosis factor biosynthetic process, positive regulation of tumor necrosis factor production, regulation of tumor necrosis factor biosynthetic process, MyD88-dependent toll-like receptor signaling pathway, positive regulation of I-κB kinase/NF-κB signaling, regulation of I-κB kinase/NF-κB signaling, regulation of interleukin-6 production, and regulation of interleukin-8 secretion)
MAPK signaling pathway (activation of MAPK activity, regulation of MAP kinase activity, positive regulation of MAP kinase activity, positive regulation of MAPK cascade, positive regulation of phosphatidylinositol 3-kinase signaling, positive regulation of ERK1 and ERK2 cascade, regulation of ERK1 and ERK2 cascade, and positive regulation of JNK cascade)
CK20 SP vs. DN
None related to major cellular function

**Table 2 ijms-22-00628-t002:** The GO analysis results between CK5/6 positive and CK5/6 negative group.

CK5/6-Positive vs. CK5/6-Negative
Cell migration (positive regulation of cell migration, positive regulation of cell motility, and regulation of cell migration)
Immune response (positive regulation of immune response, positive regulation of inflammatory response, positive regulation of lymphocyte proliferation, regulation of immune response, regulation of immune response, B cell activation, regulation of B cell proliferation, positive regulation of T cell activation, regulation of T cell proliferation, and T cell activation)
TNF signaling pathway (positive regulation of NF-κB transcription factor activity and MyD88-dependent toll-like receptor signaling pathway)
MAPK signaling pathway (positive regulation of MAP kinase activity, positive regulation of MAPK cascade, regulation of MAP kinase activity, regulation of ERK1 and ERK2 cascade, and positive regulation of JNK cascade)

**Table 3 ijms-22-00628-t003:** The ingenuity pathway analysis results (disease and function).

CK5/6 SP vs. DP	
	None related to major cellular function
CK5/6 SP vs. CK20 SP	
upregulated in CK5/6 SP	Cancer and invasion of tumor cell lines
	Adhesion of immune cells
CK5/6 SP vs. DN	
upregulated in CK5/6 SP	Cancer, neoplasia of cells, cell movement of cancer cells, cell movement of tumor cell lines, cell movement, and migration of cells
	Lymphocyte migration, cell movement of T lymphocytes, leukocyte migration, cell movement of mononuclear leukocytes, activation of lymphocytes, proliferation of immune cells, and proliferation of lymphocytes
DP vs. CK20 SP	
upregulated in DP	Activation of leukocytes, activation of mononuclear leukocytes, and leukocyte migration
	Chemotaxis
DP vs. DN	
upregulated in DP	Cancer and activation of cells
	Cell movement, migration of cells, and binding of tumor cell lines
	Activation of lymphocytes, lymphocyte migration, cell movement of lymphocytes, immune response of cells, and inflammatory response
	I-κB kinase/NF-κB cascade
CK20 SP vs. DN	
upregulated in CK20 SP	Activation of cells

**Table 4 ijms-22-00628-t004:** Clinicopathological characteristics of patients and association with IHC-defined subgroups.

	CK5/6 SP	DP	CK20 SP	DN	Total	*p*-Value
	*n* (%)	*n* (%)	*n* (%)	*n* (%)	*n*	
Age (years)						
≤68	27 (44.3%)	7 (53.8%)	29 (41.4%)	21 (46.7%)	84	0.846
>68	34 (55.7%)	6 (46.2%)	41 (58.6%)	24 (53.3%)	105	
Gender						
Male	48 (78.7%)	13 (100.0%)	58 (82.9%)	39 (86.7%)	158	0.266
Female	13 (21.3%)	0 (0.0%)	12 (17.1%)	6 (13.3%)	31	
Nuclear grade						
Low	1 (1.6%)	0 (0.0%)	3 (4.3%)	3 (6.7%)	7	0.493
High	60 (98.4%)	13 (100.0%)	67 (95.7%)	42 (93.3%)	182	
T category						
T2	55 (90.2%)	12 (92.3%)	63 (90.0%)	42 (93.3%)	172	0.927
T3~4	6 (9.8%)	1 (7.7%)	7 (10.0%)	3 (6.7%)	17	

**Table 5 ijms-22-00628-t005:** Relationship between IHC defined subgroups with immunohistochemistry expression.

	CK5/6 SP	DP	CK20 SP	DN	Total	*p*-Value
	*n* (%)	*n* (%)	*n* (%)	*n* (%)	*n*	
CK14						
Low	28 (45.9%)	12 (92.3%)	69 (98.6%)	45 (100%)	154	<0.001
High	33 (54.1%)	1 (7.7%)	1 (1.4%)	0 (0.0%)	35	
CD44						
Low	2 (3.3%)	2 (15.4%)	64 (91.4%)	34 (75.6%)	102	<0.001
High	59 (96.7%)	11 (84.6%)	6 (8.6%)	11 (24.4%)	87	
GATA3						
Low	55 (90.2%)	3 (23.1%)	11 (15.7%)	20 (44.4%)	89	<0.001
High	6 (9.8%)	10 (76.9%)	59 (84.3%)	25 (55.6%)	100	
FOXA1						
Low	56 (91.8%)	9 (69.2%)	25 (35.7%)	30 (66.7%)	120	<0.001
High	5 (8.2%)	4 (30.8%)	45 (64.3%)	15 (33.3%)	69	

## Data Availability

Data is contained within the article or Appendix A.

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
