# Peer review of "Comprehensive Gene Expression Analyses of Immunohistochemically Defined Subgroups of Muscle-Invasive Urinary Bladder Urothelial Carcinoma"

_ijms, 2021, doi:10.3390/ijms22020628_

Round 1

Reviewer 1 Report

I found the subject of article interesting. Methodology is appropriately designed and obtained results are promising. Some points are need to be address for the sake of the manuscript value.

  1. Figure 3 is not very clear, DPI should be enhanced.
  2. Author has not mentioned about the control subjects, clarify it why author has not included in study.
  3. Why author has made group <69 years and >69 years and what is mean age of patients as author has mentioned; The age of the patients ranged from 37 to 87 years with a mean age of 68 years.
  4. What is the meaning of statistical analysis of Clinico-pathological characteristics of patients?.
  5. Why author has not included the overall survival of patients with respect to IHC-based classification, CK20 expression, Impact of CK20 expression. Author should add the overall survival of patients as he has followed the patients for longer (The follow-up period ranged from 1 to 277 months).
  6. Author should mention the Median Survival Time.
  7. Author has written as: There was no significant difference in overall survival (OS) according to IHC-based classification or IHC expression (Figure 5). However, in figures X-axis has mentioned as progression free survival. Authors should correct these issues.
  8. Author should mention the statistical test used for Table 5 in statistical analysis section.
  9. The typo errors throughout the manuscript should be corrected.
  10. References are not prepared according to author guidelines.

Reviewer 2 Report

This study evaluated the role of CK5/6 and CK20 as molecular markers for bladder cancer. Bladder urothelial carcinoma was classified into CK5/6 single-positive (SP), CK20 SP, double-positive (DP) and double negative (DN) subgroups. The results showed that CK5/6 SP and DP subgroups were enriched in cell migration, immune activation, IL6-JAK-STAT3 signaling pathway and tumor necrosis factor-α signaling via NF-κB signaling pathway signature gene. The DN subgroup showed inhibited cell movement, cell migration, and cell activation. Survival analysis showed that the CK5/6 SP subgroup was significantly associated with poor progression-free survival. The authors concluded that the CK5/6 positive subgroup exhibited high gene expression signature related to aggressive behavior and exhibited worse clinical outcome.

Comments:

  1. The theme of the study is clinically relevant. Previous studies have already demonstrated the poorer prognosis in bladder cancer associated with CK5/6 expression. While there is a lack of originality, the study provides some additional insight linking CK5/6 and cancer progression.
  2. The mention of breast cancer in the Introduction section seems irrelevant.
  3. The experimental design was appropriate to answer the research questions. The study conclusion is adequately supported by the experimental findings.
  4. The manuscript and figures are well prepared.
  5. The CK5/6 SP subgroup had the worst progression-free survival but here was no significant difference in overall survival. Could the authors try to explain this discrepancy .
